# Identifying the Equilibrium Point between Sustainability Goals and Circular Economy Practices in an Industry 4.0 Manufacturing Context Using Eco-Design

**Fernando E. Garcia-Muiña [1], Rocío González-Sánchez [1], Anna Maria Ferrari [2], Lucrezia Volpi [2], Martina Pini [2]; Cristina Siligardi [3] and Davide Settembre-Blundo [1,4,*]**

[1] Department of Business Administration (ADO), Applied Economics II and Fundaments of Economic Analysis, Rey-Juan-Carlos University, 28032 Madrid, Spain
[2] Department of Sciences and Methods for Engineering, University of Modena and Reggio Emilia, 42122 Reggio Emilia, Italy
[3] Department of Engineering "Enzo Ferrari", University of Modena and Reggio, Emilia, 41125 Modena, Italy
[4] Project Management Office, Gruppo Ceramiche Gresmalt, 41049 Sassuolo, Italy
[*] Correspondence: davide.settembre@gresmalt.it; Tel.: +39-536-867-011

**Abstract:** For manufacturing companies, the transition to circular business models (CBMs) can be hampered both by the lack of relevant data and by operational tools. Eco-design, associated with Industry 4.0 IoT (Internet of Things) technologies, can be an effective methodological approach in developing products that are consistent with the principles of the circular economy. The reason is that, in the design phase, decisions are made that can significantly influence the degree of sustainability of products during their lifecycle. Therefore, in the manufacturing environment, eco-design represents an innovative approach to include sustainability among the traditional industrial variables such as functionality, aesthetics, quality, and profit. This study aimed to test eco-design as a tool to define the equilibrium point between sustainability and circular economy in the manufacturing environment of ceramic tile production, and to demonstrate how new business opportunities can be created through evolution from a linear to a circular business model, thanks to IoT and Industry 4.0 technologies used as enabling factors. The main result of this paper was the empirical validation in a manufacturing environment of sustainability paradigms through eco-design tools and digital technologies, proposing the circular business model as an operational tool to promote the competitiveness of enterprises.

**Keywords:** eco-design; sustainability; circular economy (CE); circular business models (CBMs); Industry 4.0; industrial symbiosis; industrial district (ID); Italian ceramic industry

## 1. Introduction

Nowadays, due to the competition, companies can no longer be based only on minimizing costs, but must also use their innovative capacity to increase the environmental quality of products (Panigrahi 2017). Improving environmental performance can open up new market segments to companies that were previously unexplored. These new consumers require detailed knowledge and information about the environmental costs of what they consume and use; therefore, they are capable of enabling a product's success, one that includes the attributes of quality and design as well as sustainability—that is, a product with equal functional and aesthetic performances with as little

impact as possible on the environment and society (Ceschin and Gaziulusoy 2016). Thus, companies that want to direct their innovative capacity towards the principles of sustainability will have to adopt eco-design, which is an approach in which the environmental variable assumes strategic importance (Romli et al. 2015). In this new design approach, attention to aesthetics, functionality, and cost are integrated with assessments of the flows of energies, resources, and materials needed to manufacture and use products in order to reduce their impact on the external environment, making them sustainable also from an economic–social point of view (Lacasa et al. 2016). Sustainable product design is also the first step towards a circular economy. Eco-design considers the environmental effect that the product will have throughout its lifecycle, from production to disposal (Den Hollander et al. 2017). For this reason, it is necessary to use operational tools such as Life Cycle Assessments (LCAs), which allow for the selection of low-impact resources and technological solutions that minimize waste and favor the length of the product's lifecycle up to its disposal, so that it can be easily disassembled and recycled (Kulak et al. 2016).

This paper intends to explore the adoption of eco-design to minimize the environmental and socio-economic effects of the production of ceramic tiles, rationalizing the supply system and favoring the use of resources from local sources to reduce the incidence of transport as an element of environmental criticism. The rationalization of the formulations of ceramic bodies took place within a collaborative framework of industrial symbiosis with key suppliers and using life cycle tools (i.e., LCAs and Life Cycle Costings LCCs) to define alternative design scenarios. In addition to environmental and socio-economic sustainability, the technology was also determined by testing prototypes at the laboratory scale in order to demonstrate their industrial feasibility. The monitoring of sustainability performances during the production of the best solution obtained during the design phase will be carried out through the use of IoT technologies in an Industry 4.0 environment, which will facilitate the integration of the assessment tools with the management systems for the collection and processing of process and business data. Finally, eco-design has made it possible to update the circular business model by including strategies for creating and capturing value through the marketing of products that are more environmentally friendly.

## 2. Theoretical Background

Linear production systems, which currently dominate the global economy, have proven to be resource constrained and have a high environmental and social impact because they are fundamentally based on the extraction, manufacture, use, and disposal of end-of-life products (Nasir et al. 2017). Therefore, improving efficiency by reducing the use of resources and fossil fuels will not be enough to meet today's environmental challenges (Gusmerotti et al. 2019). Linear models are exposed to fluctuating prices and access to raw materials (for economic and geopolitical reasons) and contribute to environmental degradation by affecting ecosystem services fundamental to development. In contrast to this linear economy, the circular economy, an economic concept included in the framework of sustainable development, is becoming an increasingly attractive alternative (Schroeder et al. 2019). If resource consumption continues to increase as it has in recent years, by 2050 the world's population would need three times more materials and 70% more food (Crist et al. 2017). In the next twenty years alone, the need for water and energy will be 35–40% greater. This resource race will have a significant impact on Europe's economy, in which 40% of its total costs are due to the consumption of raw materials, compared to 20% for labor costs, and based on a commodities market in which there has been an annual price increase of 6% since 2000 (Lane 2017).

In order to identify concrete projects for a circular economy, we need to look at Europe, which is now the only region in the world that already has a roadmap on its table to start applying specific criteria and rules. The European Commission stresses that the circular economy will boost the European Union's (EU) competitiveness by protecting businesses against resource scarcity and price volatility (Domenech and Bahn-Walkowiak 2019). In this case, environmental protection, human health, innovation, and improved competitiveness are embraced to define what the European economy is expected to look like in the coming decades. The EU also points out that this new way of consuming and producing creates new business opportunities and locally appropriate jobs at all skill levels and,

thus, generates opportunities for integration and social cohesion (Ghenţa and Matei 2018). To promote this new paradigm, the EU has launched various initiatives to address, in an integrated manner, some of the major challenges arising from the environmental and competitiveness problems of European industry. The "Roadmap to a Resource-Efficient Europe", framed in the European Commission's Europe 2020 Strategy, establishes actions to stimulate the market for secondary materials and the demand for recycled materials by offering economic incentives and developing criteria to determine when waste ceases to be waste (Barbosa et al. 2017). On the other hand, the Union's Seventh General Action Programme for the Environment 2013–2020 sets as its second priority the objective to turn the Union into a low-carbon (Sugiawan et al. 2019), resource-efficient, ecological, and competitive economy, (Breure et al. 2018) capable of mitigating climate change (Cucchiella et al. 2017; D'Adamo 2018). The other major European initiative is called "An Integrated Industrial Policy for the Globalization Era". It establishes six priority lines of action, among which is a sustainable industrial, construction, and raw materials policy that promotes, among others, the development of stable recycling markets and systems for extended producer responsibility, as a means of moving towards a circular economy (Lucchese et al. 2016).

The circular economy, according to the definition given by the Ellen MacArthur Foundation, "is a generic term to define an economy designed to be able to regenerate itself". In a circular economy, material flows are of two types: biological ones, capable of being reintegrated into the biosphere, and technical ones, destined to be revalued without entering the biosphere" (Korhonen et al. 2018). The circular economy is, therefore, a system in which all activities, starting from extraction and production, are organized in such a way that someone's waste becomes a resource for someone else (Fiksel and Lal 2018). Therefore, on the basis of this definition, the circular economic model ultimately seeks to decouple global economic development from finite resource consumption (Korhonen et al. 2018). It promotes key strategic objectives, such as generating economic growth (Busu 2019), creating jobs, and reducing environmental impacts, including carbon emissions (Suárez-Eiroa et al. 2019). With the economic model and linear development, we are depleting certain natural resources, so the circular economy proposes a new model of society that uses and optimizes materials and waste, giving them a second life (Paletta 2019). Thus, the product must be designed to be reused and recycled; that is, thanks to eco-design, the first to the last piece can be reused or recycled after the end of its useful life. With the circular economy, it is a question of how to convert what, up until now, has been considered waste into new raw materials (Caruso and Gattone 2019). In addition, it is also concerned with generating employment in the context of the so-called green economy. Therefore, the circular economy proposes a radical systemic change aimed at eco-design, economy of functionality, reuse, repair, remanufacturing, and industrial symbiosis (Baldassarre et al. 2019). This approach promotes innovation and long-term resilience and enables the development of new business models (Schroeder et al. 2019).

The implementation of the new philosophy of consumption and production based on the circular economy, requires, above all, training and knowledge of the different concepts associated with it. Eco-design is a key factor in the circular economy and consists of identifying, at the very moment a product/service is projected, all the environmental effects that can occur in each of the phases of its lifecycle, in order to try to reduce them to the minimum, without detriment to their quality and applications (Sauvé et al. 2016). Eco-design must consider the basic elements that make a product saleable, ranging from its appearance or aesthetics to its function, but unlike in the outdated linear economy, it must also assess all stages of its production and distribution chain, as well as economic and commercial aspects (Kuo et al. 2016). But to speak of eco-design as a model of complete product development, we must involve other concepts that consider their environmental and social repercussions. In the design of a product or service, we begin by defining its characteristics and processes: composition, raw materials to be used, how we will manufacture it, how we will transport it, and how we will market it. But we will also think about its usefulness and functionality, its durability, and how we will manage its useful life, especially in the final phase of the cycle (Castka and Corbett 2016).

Another concept linked to circularity is that of the functional economy, the purpose of which is to privilege the use over possession and, therefore, the provision of a service rather than the sale of a good.

Compared to the linear economy, this new approach is aimed at the dematerialization of production processes seen as the only way to create value (Negrei and Istudor 2018). The functional economy wants to optimize the function of the use of goods and services, maximizing their value in the long run and minimizing the consumption of material resources and energy (Urbinati et al. 2019; Sassanelli et al. 2019).

Other more common principles of environmental management are the basis of the circular economy: reduce consumption, reuse, and recycle (the so-called three Rs of environmental management). However, if considered separately, they cannot be confused with good examples of circular economy (Ghisellini et al. 2016). Some people also prefer to use another name for this type of action: "downcycling", which can be characterized as using the remains of a product to generate others with less added value (Pires and Martinho 2019). In general, the circular economy goes beyond the relatively simple practice of recycling.

The real circular economy should establish channels of collaboration among the different companies in a supply chain to achieve more efficient results. In this regard, the concept of *industrial symbiosis* is increasingly gaining ground (Domenech et al. 2019). It is a strategy for the transfer and sharing of resources among industries in the same supply chain but belonging to different sectors, such as material waste, energy by-products, services, and capacity (Herczeg et al. 2018). Industrial symbiosis favors intermediation and innovative collaboration among companies, so that the waste produced by one of them is valued as a raw material for another (Desrochers and Szurmak 2017). The adoption and dissemination of this strategy, through appropriate instruments of relations among companies, allows to obtain significant advantages from an economic and environmental point of view, making production systems more sustainable overall (Yeo et al. 2019). The strategies of industrial symbiosis are, therefore, the basis of the effective circular economy (Zaman 2017). But for industrial symbiosis to operate, the different industrial systems present on the territory must be fully integrated, not only from the point of view of production, but also from that of waste disposal (Albino et al. 2016). One of the fundamental variables for assessing the feasibility of symbiosis from an economic point of view is the distance between the waste producer and the potential user (Marchi et al. 2017). If the cost of transporting is the same, and if their price is higher than the cost of purchasing raw materials, the circular system cannot work.

From all this, the strategic importance of the supply chain arises. In a linear economy, the supply chains are the ones that extract, use, and dispose, while in a circular economy, it is the supply chains that reduce, reuse, and recycle. In a circular economy, materials are constantly circulating in many different supply chains and never have to become waste (Bressanelli et al. 2018). The biggest logistical challenges in a circular economy are the unpredictability of the flow of materials, their low financial value, and diversity of goods properties (Batista et al. 2019). Therefore, the economic and sustainable management of the supply chain will be one of the basic capabilities of successful enterprises in a circular economy (De Angelis et al. 2018).

The circular economy's perspective is then to identify the amount of resources needed for human activities within the existing and available ones, i.e., by transforming goods that have reached the end of their useful life. Waste is considered a failure of the system and the only possible correction is to transform waste and scrap into resources (Jain et al. 2018). This innovative approach must begin with the design of the product, which must be designed to last, if possible, to be repairable, and (at the end of its lifecycle) to be broken down so that each part of it finds another use. It is precisely in the concepts of recycling, reduction, recovery, repair, and reuse, which are characteristics of the circular economy, that we can identify the link between sustainability and sustainable development (Olawumi and Chan 2018). Therefore, from a circular point of view, a system should function as a biological environment where everything is functional and everything is regenerated: the concept of waste does not exist because, in fact, waste becomes the basis for the development of other forms of life in a general framework of equilibrium. Despite this, the challenge is to identify a point of equilibrium, because the system, besides being potentially regenerative, should also be sustainable (Muñoz-Torres et al. 2018). It follows that, from a sustainable point of view, not everything that could be recycled, reduced, recovered, reopened, and reused is, in fact, sustainable in environmental, social,

and economic terms. With an inverse reasoning, we can see the circular economy as a paradigm of sustainability, i.e., an innovative socio-economic approach to implementing sustainability in real life and business (Geissdoerfer et al. 2017). In a manufacturing environment, the equilibrium between the system's regenerative potential and environmental and socio-economic sustainability can be identified through eco-design.

In order to implement the principles of circular economy in business strategies, reducing dependence on increasingly scarce and expensive natural resources and turning waste into income, it is necessary to rethink or plan the value proposition and also the way in which you approach customers (Pieroni et al. 2019). But what is in practice easy to enunciate ideally, is more difficult to put into practice. Most companies, especially SMEs (small and medium-sized enterprises), are not yet ready to take advantage of the opportunities offered by the circular economy and remain firm on the more traditional model of linear growth (Tăchiciu 2018). Therefore, companies that want to enjoy the circular benefits will have to develop new business models that are not subject to the limits of linear thinking (Zucchella and Previtali 2019). These new circular business models (CBMs), in order to be able to intercept in an innovative way the value created in the supply chain, will not only have to lead the development of processes that have less impact on the environment (eco-efficiency), but will also have to take advantage of new growth opportunities to promote radically positive changes (eco-effectiveness) capable of guiding both economies and businesses towards sustainability (Heyes et al. 2018).

An important aid for companies in designing a circular business model comes from digital technologies, big data management, and artificial intelligence, because they allow for forms of collaborative innovation in supply chains (Garza-Reyes et al. 2019). Digitization allows the recording of data produced at all stages of production, marketing, management of inputs, waste, and their constant evaluation in terms of efficiency (Parida et al. 2019). Particularly in the paradigm of Industry 4.0, we can integrate information and knowledge systems based on collaborative networks. It allows a more efficient and optimized management of value chains, as well as the use of resources (Nascimento et al. 2018). The fourth industrial revolution, driven by digitization and huge volumes of data, represents the potential to leverage circular business models, where renewable resources are consumed, stocks are kept infinitely, and waste is eliminated. This is where Industry 4.0 and the circular economy meet and empower (Okorie et al. 2018). On the one hand, the disruptive technologies of the new industry operate as triggers for circular strategies. On the other hand, the circular economic model provides a purpose for Industry 4.0 and drives its development (Tseng et al. 2018).

At the end of this introductory theoretical review, we can derive several conclusions which constitute the conceptual basis of this research:

1. The circular economy represents a regenerative economic system that must maximize the creation of the value of the goods that are produced;
2. The system ensures the durability of resources through the elimination of inputs and outputs through looping of materials and components of products;
3. The lifecycle of the product is extended, and this extension also favors the connection among different value chains in the same and similar supply chains;
4. The circular economy can, thus, become a paradigm of sustainability through the use of eco-design to find the equilibrium of the system between regeneration capacity and minimization of environmental and socio-economic effects;
5. A circular business model can reduce operating costs by strengthening relations with stakeholders (suppliers, employees, customers, institutions, territory) and stimulating competitiveness.

This study seeks to fill the gaps in the literature regarding the relationship between sustainability principles and circular economy practices by addressing the following research questions:

**RQ1.** *Can eco-design be an effective tool to predict the equilibrium point between sustainability and circular economy?*

**RQ2.** *How can the circular economy create new business opportunities that combine environmental and social benefits?*

**RQ3.** *How can IoT and Industry 4.0 technologies be effective as enabling factors for the circular economy?*

## 3. Methodology

This study, based on the work of Garcia-Muiña et al. (2018), aims to operationally apply a procedure to implement the principles of environmental, social, and economic sustainability in a manufacturing environment, carrying out some of the specifications of the circular business model designed in the above paper. In the case in point, the experiment was carried out in a ceramic tile manufacturer that is among the top 10 Italian companies in the sector.

The Italian ceramic industry represents an industrial cluster of great importance both at the national and European level, as shown by the data included in the 2018 sector statistical survey published by the Italian Association of Ceramic Manufacturers (Confindustria Ceramica 2019). In 2018, the sector consisted of 137 companies with approximately 19,700 employees who produced 415 million square meters of tiles. Also, in 2018, the total turnover of Italian ceramic companies was 5.4 billion euros, of which 4.5 billion came from exports, accounting for 85% of turnover. In 2018, investments amounted to 508.2 million euros (9.4% of annual turnover), a value that has allowed the entire industry to exceed 2 billion euros in the five-year period. Among the reasons that can explain this orientation to innovation: the opportunities provided by national policies for the transition to Industry 4.0, fully taken by companies in the sector, and the recovery of competitiveness through more advanced technologies with the modernization of plants and production lines.

The diagram in Figure 1 shows the conceptual scheme of the empirical development of research and the operational procedure in relation to the research questions previously formulated. The first step is represented by the strategic phase of eco-design, i.e., the design of products that minimize their environmental effect and provide society with greater value than has been taken away from the environment, during the entire production process.

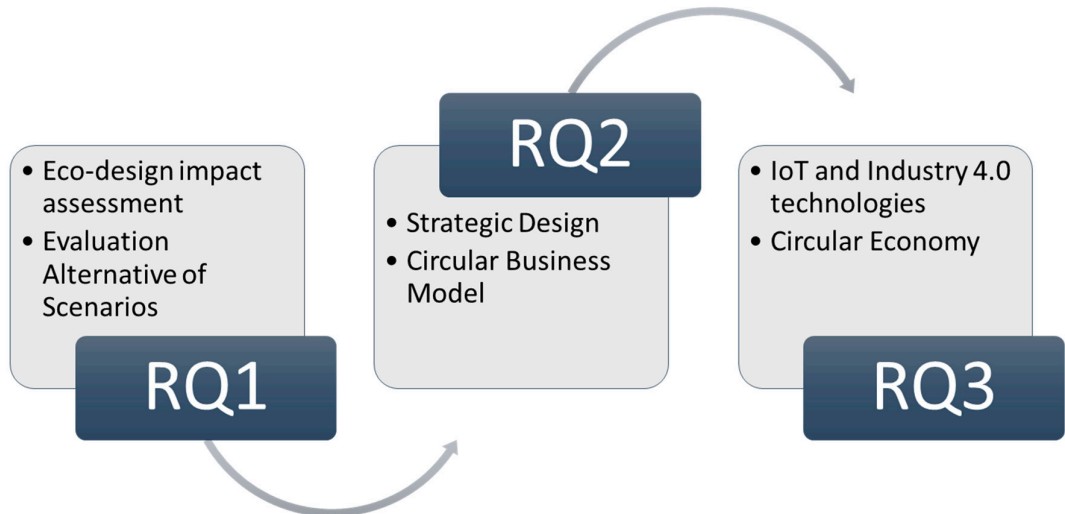

**Figure 1.** Conceptual diagram explaining the methodology adopted (RQ = research question; IoT = Internet of Things).

The Life Cycle Assessment (LCA) is used as a methodological tool to carry out eco-design (Eksi and Karaosmanoglu 2018). It allows for the entire lifecycle of the ceramic product to be assessed, quantifying the environmental effects from the sources of raw materials to manufacture, distribution, use, and final disposal. This is an internationally standardized procedure according to ISO 14040 and 14044. The LCA's logic is based on a holistic systemic approach that allows to understand and manage the complexity of the supply chain both upstream and downstream of the production process. Critical points in the entire product lifecycle are identified in order to envisage solutions aimed at saving and recovering energy and materials.

In order to take into account the socio-economic value of environmental damage in the tile manufacturing and industrial costs phases, the LCA analysis was supported by Life Cycle Costing (LCC) in order to predict the environmental and socio-economic sustainability of the different design scenarios (Lee et al. 2016). Like the LCA, the LCC also follows an international standard: ISO 15686. Both methodologies follow the scheme of four consequential phases, in accordance with the ISO standards: objective and scope, inventory analysis, impact assessment, and interpretation of results. Therefore, the research methodology was developed following exactly this logical scheme.

The second phase of the procedure was a strategic planning activity for a new circular business model and, finally, the third step involved the definition of the conceptual and operational links between the circular economy and sustainability.

## 4. Results

In a recent sustainability study carried out on a representative sample of Italian ceramic production models, it was pointed out that one of the phases of the lifecycle of the product with the greatest impact on the environment was the system of supply of raw materials (Ferrari et al. 2019). In particular, the type of transport between mines and factories (e.g., ship, train, truck) and the distance between these two locations are critical elements from an environmental point of view.

Currently, most of the raw materials used for the manufacture of ceramic tiles come from countries outside the EU—Ukraine (clays) and Turkey (feldspars). In this case, the logistics were complex; in fact, from mines, raw materials were loaded onto trucks and delivered to ports where they were shipped to Italy. Once they arrived, the materials were unloaded from ships and loaded onto trucks for transportation to factories. To a lesser extent, some clays came from Germany and, in this case, trains were used for transportation to Italy. The railway wagons arriving at the freight yard were unloaded onto trucks for delivery to factories.

Considering that about 20 kg (0.02 tons) of raw materials are needed to produce 1 square meter of tile, the Italian ceramic industry has an annual requirement of raw materials equal to:

$$415 \text{ million m}^2/\text{year} \times 0.02 \text{ tons/m}^2 = 8.3 \text{ million tons/year}$$

This figure show that the ceramic industry is a resource-intensive sector, even more so than the production process, where the transportation modes are mixed, and is a critical factor for the environment. Preliminary impact assessments (Ferrari et al. 2019) have determined that the most polluting modes of transport are ships and trucks due to their significant $CO_2$ emissions into the atmosphere. Trains, on the other hand, are the most ecological way of transport. Just as the distance between the source of supply and the factory is another factor that negatively affects the environmental impact.

### 4.1. Objective and Scope

On the assumption of this baseline, it was decided to focus the eco-design activity on optimizing the supply system to privilege more environmentally friendly transport systems, such as trains, and reducing distances between factories and mines, also using local raw materials. In order to re-engineer the ceramic material, changing the current compositional mix, it was necessary to work closely with key suppliers and other stakeholders who were represented in the same supply chain (Figure 2).

At the heart of the supply chain were manufacturers of ceramic tiles, while upstream were suppliers of materials (raw materials, inks, and glazes for decoration) and technologies (machinery). The production process was also supported by a series of ancillary service providers: graphic development studios, companies that carry out additional processing and treatments on the finished product (cutting, polishing, lapping), and suppliers of display systems for the preparation of showrooms and exhibition stands. Downstream of the manufacturers, the distribution channel was made up of various economic agents: the commercial networks of the tile manufacturers, the commercial agents external to them, and the distributors. In addition, there was another category of companies, which only carried out one commercial activity, i.e., they obtained their supplies from

ceramic manufacturers who manufacture the products they require under the brands of these companies.

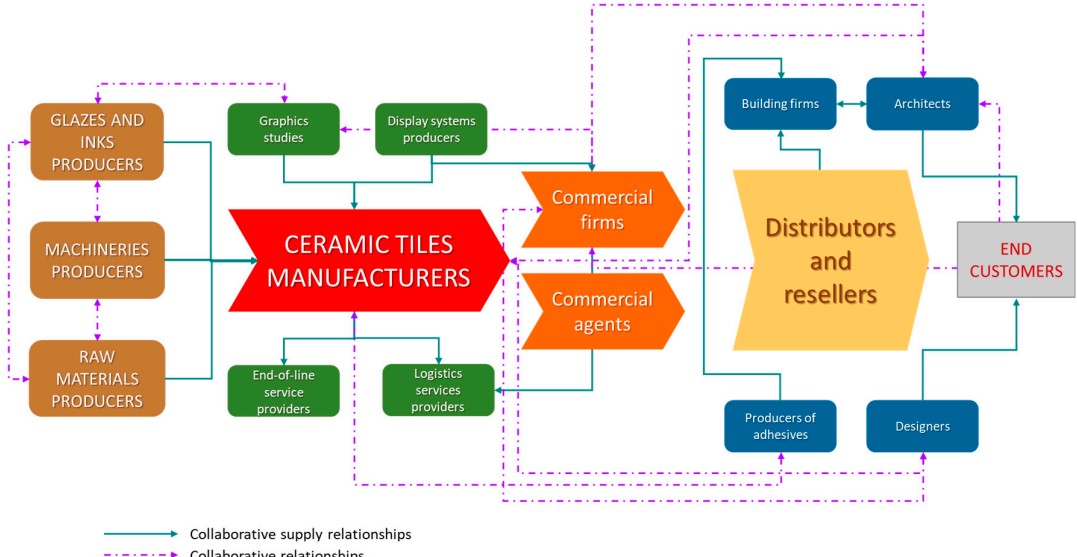

**Figure 2.** Ceramic supply chain network with collaborative relationships as the basis for industrial symbiosis.

Producers and commercial companies find themselves competing in the same markets with similar products (having shared the same technology), but mutual interest prevails: for producers to saturate production capacity by reducing industrial costs and for commercial companies to have the product to be placed on the market. From this point, the ceramic supply chain relates to the construction sector and its main economic agents: architects, designers, manufacturers of materials and solutions for the installation of floors and walls, builders, up to the end customer.

Figure 2 also shows, by means of vectors, the dynamics of collaborative relations between economic agents with and without commercial contributions for the supply of goods or services. It is clear that the supply chain is a complex system with B2B2C (business-to-business-to-consumer) characteristics, because tile manufacturers are increasingly oriented towards disintermediation of the commercial relationship by directly interacting with architects and designers overtaking distributors (Brotspies and Weinstein 2018). This relational network, typical of industrial districts, is a powerful enabling factor for industrial symbiosis and the implementation of the circular economy. Supply chain enterprises, organized in a district system, are already used to collaborate in the co-design of new technological solutions and new products. Therefore, it was decided to exploit this propensity to share knowledge to innovate the way ceramic materials are formulated, thanks to eco-design and collaboration with mining companies.

In accordance with ISO specifications for LCA and LCC analysis, 1 m$^2$ of ceramic tiles was adopted as a functional unit and the system boundaries were set from the cradle (raw materials) to the gate (end of the manufacturing process). The analysis was modelled in SimaPro®8.5.2.2 software by PRéConsultants, taking the Ecoinvent 3.4 (Wernet et al. 2016) database as a reference, especially for background processes related to natural gas, electricity, heat, transport, infrastructure, machinery, and waste treatments. The data for the impact assessment came mainly (80%) from primary sources through direct collection in the different phases of the production processes. The remaining data, on the other hand, were obtained from specialized databases.

*4.2. Inventory Analysis*

In order to implement an eco-design strategy, it is necessary to know the starting point in order to foresee alternative scenarios for environmental improvement. For this reason, a preliminary

sustainability assessment is required, starting with an inventory analysis that defines and quantifies the input and output flows in the lifecycle of the system, building a model that represents it as truthfully as possible.

First, all the phases of the lifecycle and their relationships were displayed in a process diagram, thus determining all the inputs and outputs and, therefore, the data to be collected. This scheme is shown in Figure 3, where the main production phases of the ceramic product are represented.

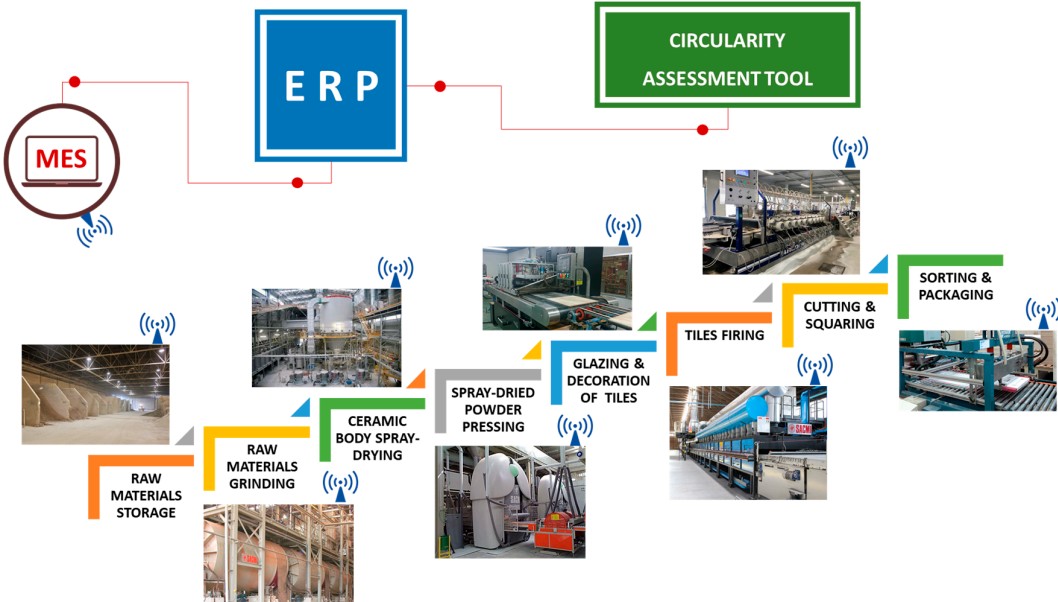

**Figure 3.** Ceramic production process layout with smart data collection system scheme.

The manufacturing process begins with the reception and storage of the raw materials that will be used to prepare the ceramic mixture. Changing the procurement and transport strategy in a radical way involves a different management of incoming flows and storage spaces. In this phase, collaboration with mining industries is of fundamental importance because, in a perspective of industrial symbiosis, it may be necessary to activate a sharing of the corresponding storage capacities to respond both to the criticality of transport and to the volatility of the demand for finished products.

After storage, the raw materials are mixed (with the compositions of the ceramic body of production) and ground with water in continuous rotary mills until a solid/liquid suspension called slip is obtained. This is then stored in underground tanks equipped with agitators. Special pumps take the slip and nebulize it inside a vertical dryer (spry-dryer), where the high pressure and high temperature cause the evaporation of the grinding water producing a very fine and homogeneous powder, ready to be pressed. During the pressing phase, the powders are dosed and transported to the hydraulic presses, which exert a pressure of over 490–500 kg/cm$^2$ on the spry-dried powder to form the support in the format (square or rectangular) and in the desired size. The pressed support is then covered with a layer of glaze and digitally decorated with special inks to obtain the required graphic design. At this point, the pressed, glazed, and decorated tiles are led to the kilns for firing at temperatures that reach 1210–1230 °C with cycles of 35–50 min depending on the size. The tiles coming out of the kiln can follow two paths: they can go directly to the packaging department of the finished product or they can be sent for further processing, which can include informed cutting of smaller and more modular tiles and/or lapping of the surface to obtain a brilliant effect such as stone materials (marble and granite).

For each phase of the process described above, data were collected on material flows, energy consumption (thermal and electrical), and emissions into the atmosphere. This procedure was implemented by exploiting the potential of IoT technologies, as the production plant analyzed was fully digitalized in line with the Industry 4.0 paradigm. As shown in Figure 2, smart meters were installed

for each machine to monitor energy consumption in real time and to collect production data. This network of sensors was wirelessly connected with the MES (manufacturing execution system), a computer system that governs and controls the entire production process, from the release of the order to the finished product, aligning the business management needs with those of the factory and, thus, bridging the gap between the decision-making level and the executive level. The MES was then integrated with the ERP (enterprise resource planning) providing real-time data on the execution of processes to allow, in addition to the current management of operations, also the inventory analysis for environmental assessment (i.e., LCA). Since the ERP system is a common and shared database of transactional data from different sources in the organization (accounting, procurement, sales, production, and logistics), it has all the information needed to carry out the inventory analysis for the economic assessment (i.e., LCC).

### 4.3. Eco-Design Impact Assessment

With eco-design, we intended to evaluate the environmental and economic behavior of alternative ceramic body compositions with respect to current production, modifying the supply strategy. The new formulations are shown in Table 1.

**Table 1.** Alternative compositional scenarios of ceramic bodies (EU = European union; P = identification code of the compositions).

| RAW MATERIALS (%) | P 01 | P 03 | P 04 | P 15 | P 17 | P 19 |
|---|---|---|---|---|---|---|
| Extra-EU clays | 25 | 25 | 10 | 5 | - | - |
| EU clays | 25 | 20 | 45 | 50 | 28 | 29 |
| Local clays | - | - | - | - | 30 | 30 |
| Extra-EU feldspar | 38 | 18 | 19 | 18 | 19 | 20 |
| Local feldspar | 5 | 19 | 10 | 24 | 10 | 11 |
| Local feldspar sand | 7 | 10 | 11 | - | 10 | 10 |
| Fired waste milled | - | 8 | 5 | 3 | 3 | - |
| Total local raw materials | 12 | 37 | 26 | 27 | 53 | 51 |

The composition P 01 was the starting point, i.e., the current production. It was characterized by a wide use of imported raw materials, 63% of which came from mines located outside the European Union and transported by ship and truck over long distances (Ukraine and Turkey, 2500–3000 km). The eco-design was, therefore, focused on three goals:

1. To minimize the use of extra-EU raw materials, favoring rail over sea and road transport;
2. To valorize local raw materials for their proximity to the factory;
3. To evaluate the possibility of using the fired waste generated during manufacture, using it as a substitute for imported feldspars by exploiting their melting properties.

Therefore, the quantity of clay from outside the EU (coming from Ukraine) was progressively reduced to the advantage of a European clay that was delivered to the factory by train from Germany. In parallel, the quantity of extra-EU feldspar (coming from Turkey) was progressively replaced with a local one (Dondi et al. 2014). In addition, quantities of fired waste were introduced on a scalar basis to verify technological feasibility and environmental impact (Table 1, compositions P 03, P 04, P 15, and P 17). The extra-EU clay was then completely removed using a large quantity of local clay (composition P 17). Finally, by comparing the compositions P 17 and P 19, it was decided to verify the environmental effect of the presence or absence of fired waste with the same composition.

Based on the inventory analysis described in Section 3.2 and considering the production process shown in Figure 2 as constant, six alternative supply scenarios were simulated, corresponding to the different body compositions indicated in Table 1. For each of them, the environmental effect was determined through a predictive LCA analysis (Hauschild et al. 2018). The results of the characterization obtained with the IMPACT 2002+ assessment method is shown in Table 2, at a mid-point level (Jolliet et al. 2003).

**Table 2.** Lifecycle assessment (LCA) for 1 m$^2$ of ceramic tiles.

| IMPACT CATEGORIES | Unit | Alternative Scenarios | | | | | |
|---|---|---|---|---|---|---|---|
| | | P 01 | P 03 | P 04 | P 15 | P 17 | P 19 |
| Carcinogens | kg C$_2$H$_3$Cl$_{-eq}$ | $4.55 \times 10^{-1}$ | $4.50 \times 10^{-1}$ | $4.45 \times 10^{-1}$ | $4.45 \times 10^{-1}$ | $4.42 \times 10^{-1}$ | $4.43 \times 10^{-1}$ |
| Non-carcinogens | kg C$_2$H$_3$Cl$_{-eq}$ | $1.23 \times 10^{-1}$ | $1.21 \times 10^{-1}$ | $1.18 \times 10^{-1}$ | $1.18 \times 10^{-1}$ | $1.16 \times 10^{-1}$ | $1.17 \times 10^{-1}$ |
| Respiratory inorganics | kg PM$_{2.5-eq}$ | $8.27 \times 10^{-3}$ | $7.47 \times 10^{-3}$ | $6.67 \times 10^{-3}$ | $6.44 \times 10^{-3}$ | $6.28 \times 10^{-3}$ | $6.36 \times 10^{-3}$ |
| Ionizing radiation | Bq C-14$_{-eq}$ | $3.93 \times 10^{1}$ | $3.66 \times 10^{1}$ | $3.34 \times 10^{1}$ | $3.27 \times 10^{1}$ | $3.17 \times 10^{1}$ | $3.21 \times 10^{1}$ |
| Ozone layer depletion | kg CFC-11$_{-eq}$ | $1.32 \times 10^{-6}$ | $1.27 \times 10^{-6}$ | $1.21 \times 10^{-6}$ | $1.19 \times 10^{-6}$ | $1.17 \times 10^{-6}$ | $1.18 \times 10^{-6}$ |
| Respiratory organics | kg C$_2$H$_4-eq}$ | $3.80 \times 10^{-3}$ | $3.58 \times 10^{-3}$ | $3.36 \times 10^{-3}$ | $3.31 \times 10^{-3}$ | $3.26 \times 10^{-3}$ | $3.29 \times 10^{-3}$ |
| Aquatic ecotoxicity | kg TEG water | $5.68 \times 10^{2}$ | $5.55 \times 10^{2}$ | $5.37 \times 10^{2}$ | $5.33 \times 10^{2}$ | $5.25 \times 10^{2}$ | $5.28 \times 10^{2}$ |
| Terrestrial ecotoxicity | kg TEG soil | $1.41 \times 10^{2}$ | $1.41 \times 10^{2}$ | $1.35 \times 10^{2}$ | $1.35 \times 10^{2}$ | $1.30 \times 10^{2}$ | $1.32 \times 10^{2}$ |
| Terrestrial acid/nutri | kg SO$_{2-eq}$ | $1.78 \times 10^{-1}$ | $1.54 \times 10^{-1}$ | $1.31 \times 10^{-1}$ | $1.23 \times 10^{-1}$ | $1.20 \times 10^{-1}$ | $1.22 \times 10^{-1}$ |
| Land occupation | m$^2$org.arable | $5.14 \times 10^{-1}$ | $4.69 \times 10^{-1}$ | $4.21 \times 10^{-1}$ | $4.00 \times 10^{-1}$ | $3.89 \times 10^{-1}$ | $3.94 \times 10^{-1}$ |
| Aquatic acidification | kg SO$_2$ eq | $3.32 \times 10^{-2}$ | $2.95 \times 10^{-2}$ | $2.61 \times 10^{-2}$ | $2.50 \times 10^{-2}$ | $2.44 \times 10^{-2}$ | $2.47 \times 10^{-2}$ |
| Aquatic eutrophication | kg PO$_4$ P-lim | $7.93 \times 10^{-4}$ | $7.51 \times 10^{-4}$ | $7.08 \times 10^{-4}$ | $7.00 \times 10^{-4}$ | $6.87 \times 10^{-4}$ | $6.93 \times 10^{-4}$ |
| Global warming | kg CO$_{2-eq}$ | 7.17 | 6.81 | 6.40 | 6.31 | 6.18 | 6.23 |
| Non-renewable energy | MJ primary | $1.78 \times 10^{2}$ | $1.73 \times 10^{2}$ | $1.67 \times 10^{2}$ | $1.66 \times 10^{2}$ | $1.64 \times 10^{2}$ | $1.64 \times 10^{2}$ |
| Mineral extraction | MJ surplus | 5.27 | 5.11 | 4.95 | 4.89 | 4.88 | 4.89 |
| Renewable energy | MJ | 4.69 | 4.57 | 4.44 | 4.42 | 4.38 | 4.40 |
| Non-carcinogens, indoor | kg C$_2$H$_3$Cl$_{-eq}$ | $1.46 \times 10^{-9}$ | $1.46 \times 10^{-9}$ | $1.46 \times 10^{-9}$ | $1.46 \times 10^{-9}$ | $1.46 \times 10^{-9}$ | $1.46 \times 10^{-9}$ |
| Respiratory organics, indoor | kg C$_2$H$_{4-eq}$ | $3.67 \times 10^{-10}$ | $3.67 \times 10^{-10}$ | $3.67 \times 10^{-10}$ | $3.67 \times 10^{-10}$ | $3.67 \times 10^{-10}$ | $3.67 \times 10^{-10}$ |
| Respiratory inorganics, indoor | kg PM$_{2.5-eq}$ | $5.01 \times 10^{-11}$ | $5.01 \times 10^{-11}$ | $5.01 \times 10^{-11}$ | $5.01 \times 10^{-11}$ | $5.01 \times 10^{-11}$ | $5.01 \times 10^{-11}$ |
| Carcinogens, indoor | kg C$_2$H$_3$Cl$_{-eq}$ | $3.76 \times 10^{-8}$ | $3.76 \times 10^{-8}$ | $3.76 \times 10^{-8}$ | $3.76 \times 10^{-8}$ | $3.76 \times 10^{-8}$ | $3.76 \times 10^{-8}$ |
| Non-carcinogens, local | kg C$_2$H$_3$Cl$_{-eq}$ | $9.11 \times 10^{-3}$ | $9.11 \times 10^{-3}$ | $9.11 \times 10^{-3}$ | $9.11 \times 10^{-3}$ | $9.11 \times 10^{-3}$ | $9.11 \times 10^{-3}$ |
| Carcinogens, local | kg C$_2$H$_3$Cl$_{-eq}$ | $2.35 \times 10^{-1}$ | $2.35 \times 10^{-1}$ | $2.35 \times 10^{-1}$ | $2.35 \times 10^{-1}$ | $2.35 \times 10^{-1}$ | $2.35 \times 10^{-1}$ |
| Respiratory organics, local | kg C$_2$H$_{4-eq}$ | $2.54 \times 10^{-3}$ | $2.54 \times 10^{-3}$ | $2.54 \times 10^{-3}$ | $2.54 \times 10^{-3}$ | $2.54 \times 10^{-3}$ | $2.54 \times 10^{-3}$ |
| Respiratory inorganics, local | kg PM$_{2.5-eq}$ | $3.13 \times 10^{-4}$ | $3.13 \times 10^{-4}$ | $3.13 \times 10^{-4}$ | $3.13 \times 10^{-4}$ | $3.13 \times 10^{-4}$ | $3.13 \times 10^{-4}$ |

The results highlight that the P 01 composition showed the highest effects in almost all impact categories, as clearly demonstrated by the highest value of each index. In particular, for the respiratory inorganics impact category, which refers to respiratory effects caused by inorganic substances, the impact was 31.7% higher than for composition P 17, which showed a lower impact; this was mainly caused by the emissions of nitrogen oxides in the air, especially due to the transport of raw materials by barge. Similarly, for the land occupation impact category, which takes into account the occupation of the soil, the impact related to composition P 01 was 32.1% higher than for composition P 17, primarily due to the land occupation related to the building for the extraction of the clay, for which the amount changed among the different compositions.

Moreover, for the aquatic eutrophication impact category, which refers to an abundance of nutrients in the aquatic environment, in particular nitrates and phosphates, the impact related to composition P 01 was 15.4% higher than for composition P 17, especially due to the emissions of phosphate in water caused by the treatment of sulfidic tailings coming from the manufacturing of the building for the extraction of the clay, for which the amount also varied. Finally, with regard to the global warming impact category, which considers the effects of greenhouse gases, the impact related to composition P 01 was 16.1% higher than for composition P 17, in particular due to the carbon dioxide emissions resulting from the transport by barge of raw materials.

Economic sustainability was assessed with the Life Cycle Costing (LCC) tool, which determines all the costs that a product generates during its lifecycle (Ciroth et al. 2015). The calculation was carried out in two phases (Andersson et al. 2016). In the first phase, we determined the economic costs attributable to the environmental effects generated by the product over its entire lifecycle (Table 2, above).

In this case, the economic value of externalities was determined, i.e., the environmental costs that the company should internalize in its industrial costs. This procedure is often referred to as Environmental LCC (E-LCC). The second phase, on the other hand, considered the industrial costs of the different phases of the lifecycle incurred exclusively by the company to manufacture the ceramic tiles (Table 2, below). This approach is referred to as Conventional LCC (C-LCC).

Regarding E-LCC, the calculation method EPS 2015dx (Steen 1999) determines the economic value of pollutant emissions based on the principle of the willingness to pay (WTP) by the polluter, both to

remedy the damage caused and to avoid further deterioration compared to the situation created. The method identifies six main categories of damage: ecosystem services, access to water, biodiversity, building technology, human health, and abiotic resources.

Clearly, the results of the environmental impacts determined by the LCA are also reflected in the environmental externalities (E-LCC). Again, the most relevant factor in terms of external costs was the distance of the mines from the factory and the transport system used. As shown in Table 2 (above), there was a progressive decrease in externalities as the quantity of local raw materials increased and rail transport was used (from composition P 01 to composition P 19).

The structure of industrial costs remained broadly unchanged in the different compositions, except for the cost of raw materials, as shown in Table 3 below. The item "cost of raw materials" includes the cost of the different materials used and the corresponding transport cost. Therefore, the greater the distance between the source of supply and the factory, the greater the cost. Furthermore, for compositions containing fired waste (P 03, P 04, P 15, and P 17), their additional grinding costs must be considered. In fact, in order to be able to use them as a partial replacement for a raw material, it is necessary to reduce their size in powder form, because they are particularly hard materials and, therefore, inadequate for reintroduction as they are into the process.

**Table 3.** Life Cycle Costing (LCC) for 1 $m^2$ of ceramic tile.

| ENVIRONMENTAL LCC (E-LCC) | | | | | | | |
|---|---|---|---|---|---|---|---|
| **DAMAGE CATEGORIES** | **Unit** | **Alternative Scenarios** | | | | | |
| | | **P 01** | **P 03** | **P 04** | **P 15** | **P 17** | **P 19** |
| Ecosystem services | €/$m^2$ | $3.19 \times 10^{-2}$ | $3.02 \times 10^{-2}$ | $2.84 \times 10^{-2}$ | $2.80 \times 10^{-2}$ | $2.74 \times 10^{-2}$ | $2.76 \times 10^{-2}$ |
| Access to water | €/$m^2$ | $1.88 \times 10^{-3}$ | $1.79 \times 10^{-3}$ | $1.69 \times 10^{-3}$ | $1.67 \times 10^{-3}$ | $1.64 \times 10^{-3}$ | $1.65 \times 10^{-3}$ |
| Biodiversity | €/$m^2$ | $1.03 \times 10^{-4}$ | $9.75 \times 10^{-5}$ | $9.17 \times 10^{-5}$ | $9.03 \times 10^{-5}$ | $8.85 \times 10^{-5}$ | $8.92 \times 10^{-5}$ |
| Building technology | €/$m^2$ | $2.80 \times 10^{-4}$ | $2.66 \times 10^{-4}$ | $2.52 \times 10^{-4}$ | $2.49 \times 10^{-4}$ | $2.44 \times 10^{-4}$ | $2.46 \times 10^{-4}$ |
| Human health | €/$m^2$ | 1.32 | 1.25 | 1.17 | 1.15 | 1.13 | 1.14 |
| Abiotic resources | €/$m^2$ | 3.14 | 3.07 | 2.98 | 2.98 | 2.92 | 2.94 |
| TOTAL (€/$m^2$) | | 4.49 | 4.35 | 4.18 | 4.16 | 4.08 | 4.11 |
| CONVENTIONAL LCC (C-LCC) | | | | | | | |
| **COST ITEMS** | **Unit** | **Alternative Scenarios** | | | | | |
| | | **P 01** | **P 03** | **P 04** | **P 15** | **P 17** | **P 19** |
| Raw materials | €/$m^2$ | 1.81 | 1.77 | 1.64 | 1.53 | 1.37 | 1.29 |
| Electrical energy | €/$m^2$ | 0.34 | 0.34 | 0.34 | 0.34 | 0.34 | 0.34 |
| Thermal energy | €/$m^2$ | 0.57 | 0.57 | 0.57 | 0.57 | 0.57 | 0.57 |
| Consumables | €/$m^2$ | 0.75 | 0.75 | 0.75 | 0.75 | 0.75 | 0.75 |
| Packages | €/$m^2$ | 0.28 | 0.28 | 0.28 | 0.28 | 0.28 | 0.28 |
| Human resources | €/$m^2$ | 1.45 | 1.45 | 1.45 | 1.45 | 1.45 | 1.45 |
| Accessories | €/$m^2$ | 1.09 | 1.09 | 1.09 | 1.09 | 1.09 | 1.09 |
| Amortizations | €/$m^2$ | 0.56 | 0.56 | 0.56 | 0.56 | 0.56 | 0.56 |
| TOTAL (€/$m^2$) | | 6.85 | 6.81 | 6.68 | 6.57 | 6.41 | 6.33 |

In this study, the social dimension of sustainability was determined through the approach of the Societal Life Cycle Costing (S-LCC), which prescribes the sum of environmental externalities (E-LCC) with industrial costs (C-LCC) (De Menna et al. 2018). This approach, compared to other methodologies such as S-LCA (Petti et al. 2018), allows to directly correlate a social indicator to the functional unit, which, in our case, corresponded to 1 $m^2$ of ceramic tile (Table 4). It is particularly important to maintain the focus of the analysis on the manufacturing process by following the cycles and times in a more dynamic time horizon than the classic social analysis, as required by the guidelines (2009) of United Nations Environment Programme (UNEP) and the Society of Environmental Toxicology & Chemistry (SETAC).

The S-LCC showed that the use of local raw materials and more environmentally friendly transport systems such as rail have a significant and positive socio-economic impact, especially when comparing the extreme compositions P 01 and P 19. In addition, it was also shown that recycling of

processing waste was not as effective in mitigating impacts as were variations in body compositions. The sustainability assessment indicated that P 17 and P 19 were the most performing compositions from an environmental point of view, with almost equivalent impact results. However, the economic analysis showed that recycling of fired waste was not beneficial for the higher incidence of pre-grinding costs. Therefore, the best result was the composition P 19.

**Table 4.** Societal LCC for 1 m$^2$ of ceramic tiles.

|  | **P 01** | **P 03** | **P 04** | **P 15** | **P 17** | **P 19** |
|---|---|---|---|---|---|---|
| Environmental LCC (€/m$^2$) | 4.49 | 4.35 | 4.18 | 4.16 | 4.08 | 4.11 |
| Conventional LCC (€/m$^2$) | 6.85 | 6.81 | 6.68 | 6.57 | 6.41 | 6.33 |
| Societal LCC (€/m$^2$) | 11.34 | 11.16 | 10.86 | 10.73 | 10.49 | 10.44 |

*4.4. Interpretation and Discussion of the Results*

Table 5 shows the main indicators of environmental, socio-economic, and technological sustainability, with eco-design for the various ceramic body compositions. In particular, the best result obtained in comparison with the starting point is highlighted.

The LCA study confirmed that the distance of the sources of supply from the factory and the type of transport used were potentially critical variables for the effects that they can generate on the environment. The scenarios defined with eco-design have indicated possible alternatives to the composition of the current ceramic body, which are more respectful of the environment.

**Table 5.** Overview of environmental, socio-economic, and technological sustainability indicators.

| **SUSTAINABILITY INDICATORS** | **P 01 Starting Point** | **P 03** | **P 04** | **P 15** | **P 17** | **P 19 Best Solution** |
|---|---|---|---|---|---|---|
| Local raw materials (%) | 12 | 37 | 26 | 27 | 53 | 51 |
| Fired waste milled (%) |  | 8 | 5 | 3 | 3 |  |
| **ENVIRONMENTAL SUSTAINABILITY** | | | | | | |
| Respiratory inorganics (kg PM$_{2.5\text{-eq}}$) | $8.27 \times 10^{-3}$ | $7.47 \times 10^{-3}$ | $6.67 \times 10^{-3}$ | $6.44 \times 10^{-3}$ | $6.28 \times 10^{-3}$ | $6.36 \times 10^{-3}$ |
| Land occupation (m$^2$org.arable) | $5.14 \times 10^{-1}$ | $4.69 \times 10^{-1}$ | $4.21 \times 10^{-1}$ | $4.00 \times 10^{-1}$ | $3.89 \times 10^{-1}$ | $3.94 \times 10^{-1}$ |
| Aquatic eutrophication (kg PO$_4$ P-lim) | $7.93 \times 10^{-4}$ | $7.51 \times 10^{-4}$ | $7.08 \times 10^{-4}$ | $7.00 \times 10^{-4}$ | $6.87 \times 10^{-4}$ | $6.93 \times 10^{-4}$ |
| Global warming (kg CO$_{2\text{-eq}}$) | 7.17 | 6.81 | 6.40 | 6.31 | 6.18 | 6.23 |
| **SOCIO-ECONOMIC SUSTAINABILITY** | | | | | | |
| Environmental LCC (€/m$^2$) | 4.49 | 4.35 | 4.18 | 4.16 | 4.08 | 4.11 |
| Conventional LCC (€/m$^2$) | 6.85 | 6.81 | 6.68 | 6.57 | 6.41 | 6.33 |
| Societal LCC (€/m$^2$) | 11.34 | 11.16 | 10.86 | 10.73 | 10.49 | 10.44 |
| **TECHNOLOGICAL SUSTAINABILITY** | | | | | | |
| Dimensional quality (ISO 10545-2) | Conform | Not Conform | Not Conform | Not Conform | Conform | Conform |
| Water absorption (ISO 10545-3) | Conform | Not Conform | Conform | Not Conform | Conform | Conform |
| Bending strength (ISO 10545-4) | Conform | Not Conform | Conform | Not Conform | Conform | Conform |

Compositional changes showed that it is possible to significantly reduce emissions into the atmosphere of particulate matter resulting from the combustion of fossil fuels that emit aerosols, sulphates, and nitrates and that can cause respiratory difficulties (impact category: respiratory inorganics). Emissions of nitrogen-containing pollutants into the environment also contribute to eutrophication, i.e., an overabundance of nitrates in water systems. This causes algae blooms that deplete the oxygen dissolved in water, consequently suffocating aquatic life (impact category: aquatic eutrophication). Similarly, new compositions of ceramic bodies can reduce the amount of carbon

dioxide ($CO_2$) released into the atmosphere due to the combustion of fossil fuels related to truck and ship transport systems. Carbon dioxide and other greenhouse gases accumulate in the atmosphere, trapping solar heat which, in turn, increases the average temperature of the Earth causing the retreat of the glaciers, the extinction of species, the loss of soil moisture, and more extreme weather conditions (impact category: global warming). A further benefit was also obtained in the category of land occupation damage, thanks to the significant use of local raw materials that use less complex mining facilities.

Furthermore, by modifying the composition of the ceramic bodies and the transport mix to maximize the use of local raw materials, reducing the distances between mines and the factory and by favoring rail transport, it was possible to estimate a reduction in externalities (E-LCC) of about 9%, comparing the initial composition (P 01) with the best result obtained (P 19). Regarding industrial costs (C-LCC), it can be noted that the introduction of fired waste as a substitute for extra-EU feldspar was not economically viable, because the externality benefit was offset by the cost of grinding the waste. Comparing the P 01 and P 19 compositions, the advantage in terms of industrial costs was always 9%, with a comparable benefit (9%) also for societal costs (S-LCC).

However, eco-design must not only be limited to the prediction of the environmental and socio-economic performance of alternative compositional scenarios but must also assess the technical and industrial feasibility of these options. We could, therefore, speak of technological sustainability to indicate that a solution complies with a set of internal specifications and/or international quality standards. For this very reason, the compositions designed were tested at the laboratory level to verify their compliance with three international standards in force for ceramic tiles. In particular, the value of water absorption (ISO 10545-3) that determines the degree of porosity of the ceramic product, the dimensions (ISO 10545-2) that establish the geometric conformity of the tiles, and the resistance to bending, which measures their mechanical properties (ISO 10545-4). The results of this further evaluation are shown in Table 5, from which it can be seen that compositional solutions with better sustainability performance than the starting point are not always manufacturable in compliance with the regulations in force.

## 5. Conclusions

With this paper, a managerial example of the introduction of the circular economy paradigm in business operations was provided. In particular, it intended to redesign the business model of a ceramic tile manufacturer through the approach of eco-design in order to optimize the supply system of raw materials to improve the environmental and socio-economic performance of the finished product.

Eco-design served to provide alternative compositional scenarios demonstrating how much the distance of the sources of supply from the factory and the transport systems used can affect the environmental and socio-economic performance of the company. With the same design approach, a feasibility study was also carried out to recycle the fired waste generated during the production process. The assessment showed that it is possible to optimize the compositions in conjunction with the company objectives in terms of sustainability. The study made it possible to identify a composition of ceramic body that performed particularly well from an environmental and socio-economic point of view, compared to the current production. This result was achieved thanks to a radical change in the composition: raw materials from outside the EU were replaced by others from local mines. This made it possible to reduce the negative impact of road transport on the environment.

The assessment also showed that recycling waste was not always beneficial from a sustainability perspective. In fact, the cost of grinding the waste baked to be used in the manufacture of tiles offset the benefit of lower external costs obtained through the recycling of waste This result can only be considered as apparently negative. In fact, it shows that the adoption of the circular economy paradigm requires a rigorous management approach, technical skills and effective tools to quantify the effects. Only with these premises can the redefinition of the business model be effective because it will have the support of a strong scientific basis.

These results provided a positive answer to the QR1 research question: eco-design is, therefore, an effective tool to predict the equilibrium point between sustainability and circular economy.

The predictive sustainability assessment was then validated at the laboratory scale in order to verify whether the new compositions conformed to the technical specifications established by the international standards for ceramic tiles. The aim was to identify this technical conformity as technological sustainability, a further dimension of sustainability which, alongside the environment, economy, and society, aims to demonstrate that the design scenario is industrially feasible. Thanks to eco-design, it was, therefore, possible to innovate the way raw materials are supplied and the industrial symbiosis within the supply chain made it possible to rationalize and make a new ceramic composition feasible. It will be the basis for the development of a finished product with better performance from the point of view of sustainability, capable of satisfying the demand for greener building products.

The development conducted in this research led to an update of the circular business model already defined previously (Garcia-Muiña et al. 2018); the changes are shown in the diagram in Figure 4 using the business model canvas (Joyce and Paquin 2016). As a value proposition, the integration between IoT technologies and sustainability monitoring systems was highlighted in order to better exploit local resources and to innovate organizational models. Networking activities were added to the key activities, performed mainly at the level of the industrial district in a framework of industrial symbiosis (Morales et al. 2019; Fraccascia et al. 2016) in order to involve raw material suppliers in a cooperative way (who play both the role of key partners and key stakeholders) in the production of products that are more environmentally friendly thanks to the use of efficient and digitized production units. These products are aimed at new market segments, such as green consumers, architects, and designers, who are more sensitive to the socially responsible behavior of the industry, also using innovative distribution channels such as digital ones. The higher costs incurred in internalizing environmental and social externalities will be offset by lower production costs and an improved reputation among stakeholders. These conclusions answer the *QR2* research question: the way to create new business opportunities by intercepting the value they generate is to prepare a circular business model that replaces the linear one (Antikainen et al. 2015).

Another important result of this study was the demonstration of the effectiveness of the Industry 4.0 paradigm as an enabling factor for sustainability, thus satisfying the *QR3* research question. In fact, thanks to the complete digitization of manufacturing, environmental, socio-economic, and technological monitoring can be carried out dynamically and in real time. On the contrary, in a non-digitized environment, sustainability assessments are conducted retrospectively based on historical datasets. There were, therefore, two innovative aspects:

1.  Eco-design, in a simulation environment, allows to predict the environmental, socio-economic, and technological performance of alternative industrial solutions;
2.  IoT technologies, in an Industry 4.0 environment, allow real-time measurement of effects as they occur, providing the capability to intervene on processes to mitigate them.

The predictive function of eco-design and the dynamic potential of digital assessment transform conventional sustainability analyses from purely technical activities to effective strategies of corporate social responsibility because the managerial perspective is changed from short to long term. The joint use of both these good practices offers the opportunity for decision-makers in manufacturing companies to apply, in a real and effective way, the principles of the circular economy by redesigning the business model and changing the way in which value is created and intercepted.

The high level of complexity reached by industrial systems requires increasingly transversal skills for an even more accurate understanding of reality from both a technological and social point of view. The advent of the fourth industrial revolution and the diffusion of digital technologies have led to the end of a world made up of silos of skills that struggled to integrate with other universes. To stimulate innovation, a multidisciplinary approach is fundamental, as was demonstrated in this research. Integrating the socio-economic dimension of sustainability with the environmental dimension required a contamination of knowledge: materials sciences, chemistry, process engineering, information technology, business organization, and management. (Table 5.)

**Table 5.** Representation of the updated circular business model inspired by the business model canvas.

| CIRCULAR BUSINESS MODEL | | | | |
|---|---|---|---|---|
| **KEY PARTNERSHIPS** | **KEY ACTIVITIES** | **VALUE PROPOSITION** | **CUSTOMER RELATIONSHIPS** | **CUSTOMER SEGMENTS** |
| Raw material suppliers<br>Suppliers of glazes and inks<br>Plant and machinery suppliers<br>Suppliers of electricity<br>Suppliers of methane<br>Packaging suppliers<br>Suppliers of chemical additives<br>IT Solution Providers<br>Financial services providers | Ceramic tile designs<br>Manufacturing of ceramic tiles<br>Marketing and sales<br>Facilities operations & maintenance<br>Sourcing<br>Logistics planning<br>Management Accounting & Control<br>Industrial Symbiosis Networking | Provide collections of porcelain stoneware tiles totally made in Italy and with the best value for money<br><br>Apply eco-design techniques to the development of new products, using ecofriendly and resource saving raw materials | Extensive sales network<br>1to1 interaction with distributors<br>Offer of ancillary services to the product<br>On-demand product development | Residential customers<br>Commercial buildings<br>Public buildings<br>Business customer<br>Green consumers<br>Architects and Designers |
| **KEY STAKEHOLDERS** | | To develop digital solutions for our manufacturing processes able to monitor in real time the environmental, socio-economic and technological performances | | |
| Private business<br>Trade channel operators<br>Suppliers<br>Staff person<br>Final consumers<br>Competitors<br>Public Institutions<br>Environment<br>Partners<br>Trade unions<br>Public and private organizations<br>Media | **KEY RESOURCES**<br><br>Three manufacturing units<br>Five logistics warehouses<br>IT Infrastructure<br>Human capital<br>Operational know-how<br>Financial assets<br>4.0 energy and resource-efficient factories | To technologically valorize the local natural resources<br><br>Be ready to innovate organizational models | **DISTRIBUTION CHANNELS**<br><br>Large-scale retails<br>Independent distributors<br>Specialized stores<br>Cloud based interactive multi-channel | |
| **COSTS STRUCTURE** | | **REVENUE STREAM** | | |
| Manufacturing costs<br>Commercial costs<br>Research and development costs<br>General and administrative costs<br>Financing cost<br>Environmental costs (externalities)<br>Social costs | | Volume of sales<br>Value recovered from the use of less expensive local raw materials<br>Better reputation from stakeholders | | |

The search for an equilibrium between the degree of sustainability of alternative compositional scenarios and the corresponding potential for circularity in tile manufacturing required the definition of a decision area with multiple criteria, where several functions needed to be optimized at the same time (Caruso et al. 2017). In this study, the search for efficiency in the use of natural resources was pursued, both as a private and collective goal. The fundamental role of managerial sciences was, thus, demonstrated by examining the positive (or interpretative) dimension of environmental and socio-economic problems. With this contribution of knowledge and by supporting the engineering sciences, mainly focused on the regulatory aspects of the problems, it was possible to obtain a more exhaustive framework linking sustainability and circular economy in businesses to support decision-making processes.

**Author Contributions:** Supervision, F.E.G.-M.; Conceptualization, R.G.-S.; Data curation, A.M.F.; Formal analysis, L.V.; Methodology, M.P.; Data curation, C.S.; Writing—review & editing, D.S.-B.

**Funding:** This research was co-funded by the European Union under the LIFE Programme (LIFE16 ENV/IT/000307: LIFE Force of the Future-Forture).

**Acknowledgments:** The authors thank the editor and three anonymous reviewers for their helpful comments on this paper.

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
