# Peer review of "Identifying the Equilibrium Point between Sustainability Goals and Circular Economy Practices in an Industry 4.0 Manufacturing Context Using Eco-Design"

_socsci, doi:10.3390/socsci8080241_

Round 1

Reviewer 1 Report

You have an interesting case here. I would suggest that you rewrite the introduction and clarify your choices. In the introduction, I would like to see your main aim and research question. What is your aim and why is it important for circularity? What is the connection between sustainability and circularity? Is circularity important here at all, is one of the questions I asked. Isn't this more about sustainability in general? Perhaps the literature review could be edited as well, depending on how you formulate the introduction. Is everything needed here actually? Now it seems to be a bit distant for the case study. The conclusions should be also rewritten and the conclusions should be better justified.

I was also wandering if Sustainability would be a more suitable journal for this article.

Author Response

Dear Reviewer,

Thank you very much for your comments and appropriate suggestions, which have contributed substantially to improving the quality of the paper. Following his indications, we have integrated the introduction by clarifying the aims of the research and formulating 3 research questions in this regard. In addition, we have underlined and discussed the relationship between sustainability and circularity from the title to the conclusions. Finally, we have also updated the analysis of the literature to make it consistent with the new structure of the paper. All changes and additions are indicated in blue ink.
This research is part of an integrated project and the first results have already been published by Social Sciences, so the editor asked us to follow up our first paper of conceptual content, proposing a second of a more operational character.

The Authors.

Reviewer 2 Report

Refine abstract section, Line 12 and Line 13.

Abstract is a bit lengthy- does this meet the requirements of Social Sciences Journal? 

Line 12 and Line 25: You write, "Eco-design" and then "eco-design". Be consistent with language.

Line 36: Do you mean, "there are consumers?"

Line 37: "decree the success?" Decree does not seem right.

In your introduction I do not see a definition of eco-design. 

I am not sure you have accurately and clearly linked eco-design to the circular economy.

Line 135: Change the phrase, "some people even preferred to use"

Line 150: "systemissionms"???

Line 166: "plan ex"???

It would be good to have a kind of simple conceptual diagram to explain your methodology (and its links to the introduction and other parts of your paper)

Line 253: ton or tonnes? Check.

Figure 1: Ensure the title of Figure 1 follows with the body of the figure.

Table 1 requires its own heading. The spaces within column that is unfilled, please replace with a dash.

Table 2 should fit in a single page. At the moment it is broken.

378: How does the result highlight that the composition P01 shows the highest impacts?

Author Response

Dear reviewer,

Many thanks for the feedback and suggestions that have allowed us to significantly improve our manuscript.
The errors, misprints and recommendations you have indicated for each line have all been corrected by indicating them with blue ink in the text.
We also introduced the concept of eco-design in the introduction, contextualizing it with respect to the concepts of sustainability and circular economy. So as you suggested we introduced a new conceptual scheme that shows the development of the work (Figure 1). We believe that it significantly improves the ease of reading and understanding of the work. Finally, we have better specified the reasons why the P01 composition shows the greatest impacts.

The Authors

Reviewer 3 Report

The paper is very interesting and it is adaptable to this special issue. I think that the concept proposed in this work is extremely interesting and I suggest the authors to provide some modifications.

This work provides a concrete example of the implementation of the circular economy to an industrial reality, thanks to the use of eco-design and IoT technologies.  However, in order to exploit the results of this research, which cover a lack in both the academic and managerial fields, the paper should be revised in its introductory part to facilitate reading and possibly a more pertinent title could also be adopted.

The theoretical framework provided by the authors is sufficiently deep and well detailed, and includes the latest scientific work on sustainability and the circular economy.

However, the introduction is disconnected from the empirical development and the reader finds it difficult to understand what the actual aims of the research are.

The research design could then be substantially improved by linking the scope of the work with the results and conclusions. To this end, it would probably be useful to enunciate some research questions at the end of the introduction.

Although the subject is a technical one, the methodological approach is well presented and sufficiently clear. In this regard, it would be interesting for the authors to highlight the potential of contamination between social and empirical sciences as a new way to promote technological and organizational innovation.

Finally, the conclusions would be better supported by the results if they were linked to the aims of the work, for example by providing answers to any research questions that may be asked when introducing the paper.

Author Response

Dear Reviewer,

Thank you very much for your appreciation of our work and for your valuable suggestions to further improve it.
As you suggested we have integrated the introductory part to better define the aims of the work, for this reason three research questions have been formulated that justified the empirical development. They also allowed us to link the results with the theoretical framework and conclusions. We also highlighted and discussed the importance of a multidisciplinary approach to building a comprehensive framework of sustainability and circularity strategies. Finally, we highlighted the importance of the relationship between sustainability and the circular economy as early as the title, which has been changed. This concept has become the new thread of the paper. All changes and additions are indicated in blue ink. 

The Authors

Round 2

Reviewer 3 Report

The paper can be accepted in present form.